# BREAKING THE BIAS: QUANTIFYING THE ATTENTION OF INDUSTRIAL ANOMALY DETECTION

## ABSTRACT

Industrial anomaly detection (IAD) predominantly utilizes unsupervised learning due to the scarcity and unpredictability of defect samples. A major challenge in unsupervised IAD methods is the inherent bias in normal samples, which causes models to focus on variable regions while overlooking potential defects in invariant areas. In this paper, we propose **R**ecalibrating **A**ttention of Industrial **A**nomaly **D**etection (RAAD), which decomposes and recalibrates the input data to highlight anomalies better. Additionally, Hierarchical Quantization Scoring (HQS) is introduced to refine the detection process by assigning quantization scores at multiple levels. These strategies work together to mitigate the bias toward normal samples and improve the accuracy of anomaly detection. We validate the effectiveness of RAAD on three IAD datasets: MVTec-AD, MVTec-LOCO, and VisA. The experimental results demonstrate that RAAD exhibits competitiveness in both detection and localization tasks, providing a robust solution for industrial anomaly detection. The source code will be released to promote further research and application.

## 1 INTRODUCTION

Industrial anomaly detection (IAD) is crucial for maintaining the quality and safety of manufacturing processes. Because of high annotation costs and the unpredictable nature of defects, unsupervised methods have become a practical solution for real-world anomaly detection, which is only trained on normal samples. However, traditional unsupervised methods face a fundamental challenge: during training, models tend to overfit the changing parts of normal samples while overlooking potential defects in unchanged regions. As shown in Figure 1, we visualize this challenge through anomaly heatmaps on MVTec-AD Bergmann et al. (2019a) and MVTec LOCO Bergmann et al. (2022) datasets. Within each group, from left to right, are the normal samples of the corresponding categories, the average anomaly heatmap for normal samples, and the average anomaly heatmap for anomalous samples. Brighter areas in the heatmaps indicate regions with a higher likelihood of receiving attention. The white boxes in the second column highlight how the model is misled by the inherent bias in normal samples. In other words, the attention maps derived from unsupervised training tend to highlight variable regions in normal samples, thereby neglecting invariant regions where subtle anomalies may reside. One might intuitively consider abandoning the attention mechanism. Nevertheless, ignoring attention maps entirely is not a viable solution, as they play a crucial role in anomaly detection. A key question arises: how to make the model allocate attention more reasonably?

A feasible solution is to solve this problem with two steps: first, directing the model's attention toward the primary target, and then reallocating the attention for improved anomaly detection. The former can be achieved through model quantization, while the latter is accomplished via fine-tuning. During quantization, the reduction in parameter precision compels the model to prioritize learning and extracting the most critical information.Meanwhile, during the fine-tuning process, the model's attention is recalibrated, enabling the redistribution of attention to better align with task-specific requirements. Building on this insight, we propose RAAD (**R**ecalibrating **A**ttention of Industrial **A**nomaly **D**etection), which firstly modifies attention maps with quantization and then fine-tuning them to recalibration. Meanwhile, we observe that convolutional neural networks are commonly used as backbone networks for extracting image features in industrial anomaly detection tasks, with each layer having a different impact on the model's attention. To optimize the attention allocation process, we introduce **H**ierarchical **Q**uantization **S**coring (HQS), which

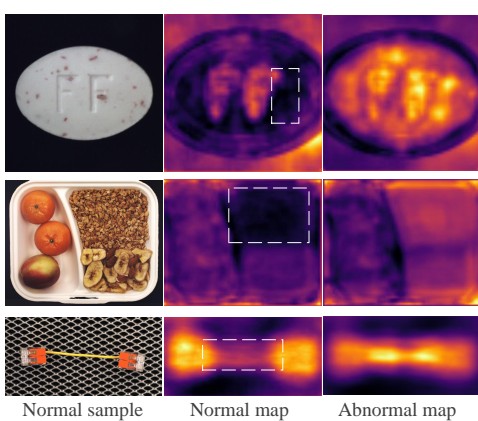

Figure 1: Visualization of heatmaps. These samples are from the MVTec-AD and MVTec LOCO datasets, which represent examples of industrial products, the average heatmap of normal samples, and the average heatmap for anomaly samples, respectively. It clearly shows the bias contained in the normal samples compared to the abnormal samples.

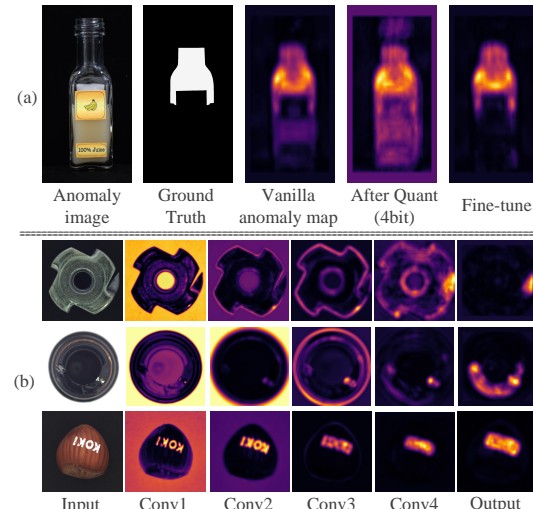

Figure 2: (a) Visualization of the attention maps at different stages of the model, from left to right, are the anomaly image, ground-truth, and predicted anomaly score. (b) The layer-wise attention outputs demonstrate the varying importance of each layer in anomaly detection.

adaptively allocates bit-width according to each layer's anomaly detection capability. In Figure 2(a), we visualize the anomaly heatmaps before and after model quantization. It can be observed that, compared to before quantization, the model's attention is more spread across the main subject while ignoring the background. Subsequently, more precise anomaly detection was achieved after fine-tuning. Figure 2(b) illustrates the outputs of each convolutional layer in the teacher-student network, highlighting the layer-wise variation in focus across the image. This design leverages the distinct roles of network layers: shallow layers capture local details, while deeper layers extract global features, and it is most beneficial for enhancing model performance with fewer parameters.

Our main contributions are as follows: We break the inherent bias in attention allocation within unsupervised IAD, guiding models to better detect subtle anomalies in invariant regions. We proposed RAAD, which systematically refines attention maps, using quantization to reduce bias and recalibrating the attention map via fine-tuning to improve anomaly sensitivity. We introduce HQS, a module that dynamically allocates bit-widths based on each layer's anomaly detection capability, optimizing the alignment between quantization and attention for enhanced efficiency and accuracy in IAD.

## 2 RELATED WORK

### 2.1 UNSUPERVISED INDUSTRIAL ANOMALY DETECTION.

Based on deep learning, visual detection has made significant achievements with the assistance of supervised learning, as cited in Kwon et al. (2019); Ruff et al. (2021). However, in real-world industrial scenarios, the scarcity of defect samples, the cost of annotation, and the lack of prior knowledge about defects may render supervised methods ineffective. In recent years, unsupervised anomaly detection (IAD) algorithms have been increasingly applied to industrial detection tasks, as referenced in Sydney et al. (2019); Xie et al. (2024); Tao et al. (2022). "Unsupervised" means that the training phase only includes normal images, without any defect samples. IAD refers to the task of differentiating defective images from the majority of non-defective images at the image level. Unsupervised IAD is mainly categorized into three types, i.e., the reconstruction-based methods, the synthesizing-based methods, and the embedding-based methods. Feature embedding-based methods have recently achieved state-of-the-art performance and can be specifically categorized into: teacher-student architecture Bergmann et al. (2020); Deng & Li (2022), normalizing flow Rezende &

Mohamed (2015); Rudolph et al. (2021), memory bankRoth et al. (2022); Cohen & Hoshen (2020), and one-class classification Sohn et al. (2020).

The most typical methods are the memory bank and teacher-student architecture. Memory bank methods embed normal features into a compressed space. Anomalous features are distant from the normal clusters within the embedding space. Regarding the teacher-student architecture, the teacher is a pre-trained and frozen CNN, and the student network is trained to mimic the teacher's output on training images. Since the student has not seen any anomalous images during training, it is generally unable to predict the teacher's output on these images, thereby achieving anomaly detection. Uninformed Students Bergmann et al. (2020) first introduced a new framework for anomaly detection known as the teacher-student anomaly detection framework. Reverse Distillation (RD) Deng & Li (2022) proposed a method where the student decoder learns to recover features from the compact embeddings of the teacher encoder. The GCCB Zhang et al. (2024) method employs a dual-student knowledge distillation framework, enhancing the ability to detect structural and logical anomalies. However, methods based on feature embeddings rely on the size of the memory bank or the capability of the teacher network. This reliance can lead to excessive memory usage, resulting in slower inference times, or may limit the model's generalization ability.

Reconstruction-based methods Haselmann et al. (2018); Ristea et al. (2022); Zavrtanik et al. (2021b) span from autoencoders Bergmann et al. (2019b); Zavrtanik et al. (2021a); Chen et al. (2023) and generative adversarial networks Yan et al. (2022); Duan et al. (2023) to Transformers You et al. (2022); Yao et al. (2023) and diffusion models Lu et al. (2023); Zhang et al. (2023). Among them, autoencoder methods rely on accurately reconstructing normal images and inaccurately reconstructing anomalous ones, detecting anomalies by comparing the reconstruction with the input image. Reconstruction-based methods are more likely to capture information from the entire image Liu et al. (2020).

### 2.2 QUANTIZED NEURAL NETWORKS.

Quantization aims to compress models by reducing the bit precision used to represent parameters and/or activations Cai et al. (2020). Existing neural network quantization algorithms can be divided into two categories based on their training strategy: post-training quantization (PTQ) and quantization-aware training (QAT). PTQ Nagel et al. (2019) refers to quantizing the model after training, without any fine-tuning or retraining, thus allowing for quick quantization but at the cost of reduced accuracy. In contrast, QAT Gong et al. (2019); Dong et al. (2020) adopts an online quantization strategy. This type of method utilizes the whole training dataset during the quantization process. As a result, it has higher accuracy but limited efficiency.

Recently, several studies have explored the integration of quantization techniques into anomaly detection tasks Sharmila & Nagapadma (2023). For example, Cho (2024); Jena et al. (2024) have even applied Post-Training Quantization for On-Device Anomaly Detection, striking a balance between computational efficiency and detection accuracy. However, it is important to clarify that model quantization is the method in this paper, not the goal.

Our method leverages the precision reduction characteristics of PTQ to achieve dimensionality reduction in weight precision, while also designing a mixed-precision quantization method specifically tailored for industrial anomaly detection.

## 3 METHOD

In this section, we will introduce our approach from three aspects: 1. Model architecture, which explains the details of the teacher-student model and the autoencoder we use. 2. Model quantization. How to evaluate the anomaly detection capability of each layer through the feature of the teacher-student model and determine the bit-width configuration. 3. Model training and fine-tuning methods. The specific operations for model initialization and fine-tuning.

### 3.1 MODEL ARCHITECTURE

Our model consists of a teacher-student model and an autoencoder, as illustrated in Figure 3. The RAAD process is divided into three steps: 1. Model Initialization: The model is trained on a dataset containing only normal images. During training, only the weights of the student model and the

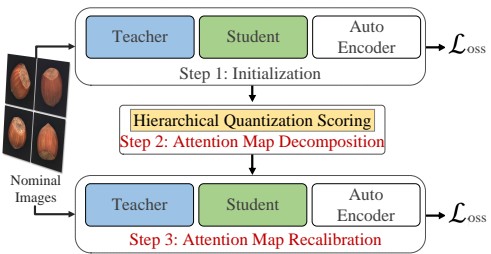

Figure 3: Pipeline of RAAD. Our architecture consists of three components: the teacher-student model and the autoencoder. During training and fine-tuning, we only use normal images. The process is divided into three steps: 1. Initial training of the model, 2. Decomposition of attention map in hierarchical quantitative scoring, detailed in Figure 4. 3. Fine-tuning of attention recalibration.

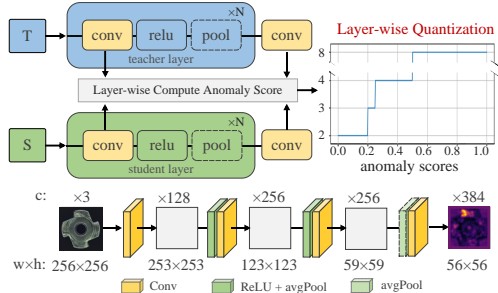

Figure 4: Hierarchical Quantization Scoring (HQS) Module. The teacher and student models are aligned layer by layer, with the anomaly scores calculated using the outputs of their respective convolutional layers. These scores are then converted into quantization bit-widths through a piecewise function. Below are the details of the teacher-student network (PDN).

autoencoder are updated. 2. Hierarchical Quantization Scoring: We first evaluate the anomaly detection capability of each layer in the network. Then, post-training quantization of the model layer by layer. 3. Attention Map Recalibration: Similar to the first step, we fine-tune the student model and the autoencoder. Before training, we employ a pre-trained WideResNet-101 (WRN-101) Zagoruyko & Komodakis (2016) on ImageNet to initialize the teacher model. By minimizing the Mean Squared Error (MSE) between the teacher model and the pre-trained network features. The loss function is as follows:

$$L_{pre} = \frac{1}{cwh}||(\Psi(I)_{c,w,h} - \mu)(\sigma) - T_{c,w,h}(I)||_F^2, \tag{1}$$

where $I$ represents an image from the ImageNet, $E$ is a feature extractor composed of the second and third layers of the pre-trained WRN-101 network, and $T(\cdot)$ refers to the teacher model. $\mu$ and $\sigma$ are the mean and standard of the $\Psi(I)$. $||\cdot||_F$ denotes the Frobenius norm (i.e., the square root of the sum of the squares of all elements).

We utilize the Patch Description Network (PDN)Batzner et al. (2024) as both the teacher and student model's feature extraction network. Unlike recent anomaly detection methods that commonly employ pre-trained CNN networks, such as DenseNet-201Li et al. (2023); Huang et al. (2017) and WideResNet-101 Esser et al. (2019); Zagoruyko & Komodakis (2016), the PDN consists of only four convolutional layers. It is fully convolutional and can be applied to images of variable sizes. As the network depth and parameters are reduced, the running time and memory requirements of the model are correspondingly reduced, achieving an inference speed of 113 FPS for a single image on an NVIDIA RTX3090 GPU.

## 3.2 ATTENTION MAP DECOMPOSITION

The industrial anomaly detection model is trained using a block-level reconstruction strategy Li et al. (2021), followed by layer-by-layer quantization. The quantization formula is as follows:

$$r = S(q - Z), \quad q = \text{clip}(\text{round}(\frac{r}{S} + Z), 0, 255), \tag{2}$$

where $r$ is the pre-quantized floating-point number, with the range $(r_{\min}, r_{\max})$. $q$ represents the quantized fixed-point number, with the range $(q_{\min}, q_{\max})$. $S$ is the scaling factor, given by:

$$\min_{\hat{\mathbf{w}}} \mathbb{E}\left[\Delta\mathbf{z}^{(\ell),\mathrm{T}}\mathbf{H}^{(\mathbf{z}^{(\ell)})}\Delta\mathbf{z}^{(\ell)}\right] = \min_{\hat{\mathbf{w}}} \mathbb{E}\left[\Delta\mathbf{z}^{(\ell),\mathrm{T}}\text{diag}\left(\left(\frac{\partial L}{\partial\mathbf{z}_1^{(\ell)}}\right)^2, \ldots, \left(\frac{\partial L}{\partial\mathbf{z}_a^{(\ell)}}\right)^2\right)\Delta\mathbf{z}^{(\ell)}\right], \tag{3}$$

$S = \frac{r_{\max} - r_{\min}}{q_{\max} - q_{\min}}$, and $Z$ is the zero-point, given by: $Z = \text{clip}(\text{round}(-\frac{r_{\min}}{S}), 0, 255)$, where clip is a clipping function that constrains values within the range of 0 to 255.

By using the Fisher information matrix, we measure the inter-block relationships within layers. Any second-order error difference is transformed into blockwise output, activating the Fisher information matrix's diagonal elements to match each element's variance. The weighted objective is:

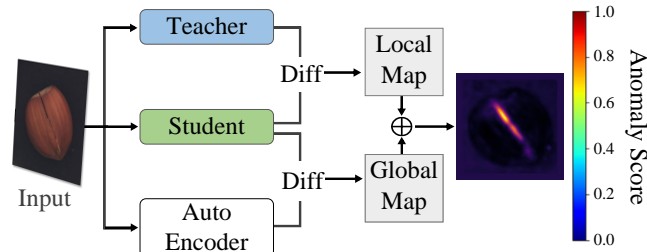

Figure 5: The inference process of the models. "Diff" refers to computing the element-wise squared difference between two collections of output feature maps and computing its average across feature maps. To obtain pixel anomaly scores, the anomaly maps are resized to match the input image using bilinear interpolation.

where $\Delta z^{(l)}$ is the feature variation in the $l$-th layer, which is the difference between quantized and pre-quantized values. $\mathbb{E}[\cdot]$ represents the expected value. $H^{(z^{(l)})}$ is the Hessian matrix of features in the $l$-th layer.

### 3.3 HIERARCHICAL QUANTIZATION SCORING

To further enhance the redistribution of attention in anomaly detection, we propose a Hierarchical Quantization Scoring (HQS) mechanism, which utilizes mixed-precision techniques to adaptively adjust the bit-width for each layer. As illustrated in Figure 4, both the teacher and student models are processed through the HQS module, where corresponding layers of the same depth are aligned, and their outputs after convolution are used to compute an anomaly score $s$ ($s \in (0, 1)$). This score measures the alignment of attention between the two models, guiding the reallocation of attention to defect-prone regions.

The bit-width $b$ for the $\ell$-th layer ($1 \le \ell \le N$) is determined as a function of its anomaly score. Layers with higher scores, which contribute more to accurate anomaly localization, are assigned greater precision, allowing deeper, semantically rich layers to focus on subtle anomalies. In contrast, shallower layers, which often capture redundant or noisy information, are more heavily compressed. This dynamic bit-width allocation leads to a more effective redistribution of attention, ensuring that the model emphasizes critical regions in the attention maps. The bit width $b$ for the $\ell$-th layer ($1 \le \ell \le N$) is defined as:

$$b^{(\ell)} = \phi(anomaly\_score) \quad = \phi\left( (c^{(\ell)} w^{(\ell)} h^{(\ell)})^{-1} \sum_c \left\| \mathrm{T}_c^{(\ell)}(i) - \mathrm{S}_c^{(\ell)}(i) \right\|_F^2 \right), \qquad (4)$$

where $i$ represents the output of the previous layer, when $\ell = 1$, $i = conv^{(1)}(I)$, with $I$ being the image input to the model. $T(i), S(i) \in \mathbb{R}^{c \times w \times h}$, where $c, w, h$ are the number of channels, width, and height of the output features of the $\ell$-th layer, respectively. $\phi(\cdot)$ is a piecewise function that determines the bit width based on the hierarchical quantization score, and the detailed explanation of the piecewise function is in the appendix. We chose 2, 3, 4, and 8 bits for mixed precision because they are most common in practical deployment.

### 3.4 ATTENTION MAP RECALIBRATION

During the training process, the teacher model, student model, and autoencoder are paired with each other to generate three losses: $L_{t-s}, L_{ae-s}, L_{t-ae}$. Formally, we apply the teacher $T$, student $S$, and autoencoder $A$ to the training image $I$, with $T(I), S(I), A(I) \in \mathbb{R}^{C \times W \times H}$, and the loss expression for $L_{t-s}$ is:

$$L_{\mathrm{t}-s} = (CWH)^{-1} \sum_c \| T(I)_c - S(I)_c \|_F^2, \qquad (5)$$

The expressions for $L_{ae-s}$ and $L_{t-ae}$ are similar to $L_{t-s}$, differing in that $T(I)_c - S(I)_c$ is replaced with $A(I)_c - S(I)_c$ and $T(I)_c - A(I)_c$, respectively. Note that to confine $L_{t-s}$ to the most relevant

Table 1: Comparison results with baseline and alternative quantification methods on MVTec-AD and MVTec-LOCO. $\cdot / \cdot / \cdot$ denotes Anomaly Image-level AU-ROC%/Image-level AP%/PRO%. Grey indicates the average value. † denotes the unofficial implementation of EfficientAD. The best results are highlighted in **bold**.

| Dataset | Baseline† | LSQ | OMPQ | RAAD |
|---|---|---|---|---|
| **Weight/Activation** | 32/32 | 8/8 | 8/8 | $\leq 8/\leq 8$ |
| **MVTec-AD** | 96.98/97.44/91.38 | 97.21/96.58/86.19 | 98.77/97.97/92.17 | **98.90/97.83/92.92** |
| bottle | **100.0/100.0/94.58** | 99.92/98.32/88.82 | **100.0/100.0**/93.92 | **100.0/100.0**/93.97 |
| cable | 95.16/ **100.0/94.58** | 95.25/86.34/87.39 | 96.49/90.72/86.80 | **97.71**/94.51/88.75 |
| capsule | 94.41/94.32/96.00 | 85.32/87.23/79.17 | 96.33/**96.40/96.78** | **97.40**/94.74/96.70 |
| carpet | 97.43 /98.91/91.11 | 98.17/98.51/90.38 | 98.71/98.84/91.09 | **98.79/98.85/91.99** |
| grid | 99.08/100.0/88.84 | **100.0/100.0**/88.85 | **100.0/100.0**/88.75 | 99.83/100.0/**91.00** |
| hazelnut | 99.50/**100.0**/91.40 | 99.14/99.18/83.88 | 99.50/**100.0/92.44** | 99.78/100.0/**92.44** |
| leather | 86.68/93.23/97.09 | 98.30/97.20/97.87 | **99.79/98.92/98.22** | 98.30/93.88/97.87 |
| metal_nut | 98.43/98.76/91.86 | 97.99/98.24/89.09 | 98.77/**98.91**/93.28 | **98.82/98.91/93.66** |
| pill | 96.78/**98.34**/95.87 | 92.03/95.33/84.14 | 97.84/97.18/97.41 | **98.00**/97.18/**97.44** |
| screw | 93.72/93.44/89.87 | 96.67/96.55/90.96 | 98.48/95.08/**94.97** | **98.56/96.64**/94.45 |
| tile | **100.0/100.0**/88.42 | **100.0/100.0**/88.42 | **100.0/100.0**/88.42 | **100.0/100.0/89.00** |
| toothbrush | **100.0/100.0**/94.47 | **100.0/100.0**/58.72 | **100.0/100.0/94.47** | **100.0/100.0/94.47** |
| transistor | 99.54/**100.0**/85.43 | 99.29/**100.0**/85.43 | **100.0/100.0**/87.31 | **100.0/100.0/87.31** |
| wood | 98.77/97.63/87.41 | **99.03**/97.69/**87.57** | 98.68/**98.36**/86.65 | 98.50/**98.36**/87.00 |
| zipper | 95.24/94.43/91.67 | 97.05/94.18/92.15 | 96.95/**95.16**/92.05 | **97.84**/94.44/**97.84** |
| **MVTec-LOCO** | 84.09/ 78.51/83.32 | 86.26/82.34/81.16 | 89.60/**88.59**/86.19 | **89.75**/87.85/**86.76** |
| breakfast_box | 77.13/ 66.98/65.21 | 80.54/66.93/65.31 | 80.91/**91.79**/72.00 | **80.94**/85.81/**72.31** |
| juice_bottle | 96.41/ 96.79/97.28 | **99.62/98.72/98.29** | 97.86/97.41/97.96 | 98.21/97.01/98.07 |
| pushpins | 78.35/ 68.12/88.23 | 78.97/85.03/83.47 | 95.46/90.59/90.86 | **95.85/91.18/91.09** |
| screw_bag | 71.57/ 67.80/76.13 | 74.65/66.25/64.25 | **76.02/67.86**/75.41 | 75.95/66.36/**77.64** |
| splicing_connectors | 97.03/ 78.51/94.53 | 97.56/94.76/94.50 | 97.75/95.29/**94.74** | **97.83/98.89**/94.72 |

parts of the image, the value of 10% is used for backpropagation in each of the three dimensions of the mean squared error $D$, where $D_{c,w,} = \left( T(I)_{c,w,h} - S(I)_{c,w,h} \right)^2$.

The total loss is the weighted summation of the three:

$$Loss = \lambda_{t-s} L_{t-s} + \lambda_{ae-s} L_{ae-s} + \lambda_{t-ae} L_{t-ae}. \qquad (6)$$

As illustrated in Figure 5, the inference process after training involves the teacher-student outputs a local anomaly map, while the autoencoder-student outputs a global anomaly map. These two anomaly maps are averaged to calculate a composite anomaly map, with its maximum value used as the image-level anomaly score, where the $2D$ anomaly score map $M \in R^{W \times H}$ is given by $M_{w,h} = C^{-1} \sum_c D_{c,w,h}$, which is the cross-channel average of $D$, assigning an anomaly score to each feature vector.

## 4 EXPERIMENT

In this section, we demonstrate the effectiveness of RAAD, by comparing the impact of different quantization methods on the performance of the model and comparing our proposed method with other advanced IAD methods. Moreover, we provide additional ablation studies.

### 4.1 DATASETS AND EVALUATION METRIC

**MVTec-AD** Bergmann et al. (2019a) dataset is a widely recognized anomaly detection benchmark that encompasses a diverse dataset of 5,354 high-resolution images from various domains. The data is divided into training and testing sets, with the training set containing 3,629 anomaly-free

Table 2: Comparison with some state-of-the-art methods. $\cdot/\cdot$ denotes image-level AUROC%, pixel-level AUROC%. Mean Anomaly Image-level AU-ROC% on MVTec-LOCO, and VisA.

| Dataset | MVTec | LOCO | VisA |
|---------|-------|------|------|
| PatchCore | 99.2/98.1 | 80.3 | 95.1 |
| SimpleNet | **99.6**/98.1 | 77.6 | 96.1 |
| EfficientAD | 96.9/97.1 | 84.0 | 95.3 |
| RealNet | **99.6**/99.0 | - | 96.3 |
| RAAD | 98.9/97.9 | **89.7** | **96.7** |

Table 3: Ablation studies on our RAAD. "Quant": The model utilizes post-training quantization, where the weights and activations are quantized to 8-bit precision, followed by fine-tuning. "HQS": Using Layer-wise mixed precision Quantization.

| Quant | HQS | I-AUC | PRO |
|-------|-----|-------|-----|
|  |  | 96.98 | 91.38 |
| ✓ |  | 98.77 | 92.17 |
| ✓ | ✓ | 98.90 | 92.92 |

images, ensuring a focus on normal samples. On the other hand, the test set consists of 1,725 images, providing a mix of both normal and abnormal samples for comprehensive evaluation. To aid in the anomaly localization evaluation, pixel-level annotations are provided. **MVTec-LOCO** Bergmann et al. (2022) dataset includes both structural and logical anomalies. It contains 3644 images from five different categories inspired by real-world industrial inspection scenarios. Structural anomalies appear as scratches, dents, or contaminations in the manufactured products. Logical anomalies violate underlying constraints, e.g., a permissible object being present in an invalid location or a required object not being present at all. **VisA** dataset Zou et al. (2022) proposes multi-instance IAD, comprising 10,821 high-resolution images, including 9,621 normal images and 1,200 anomaly images. This dataset is organized into 12 unique object classes. These 12 object classes can be further categorized into three distinct object types: Complex Structures, Multiple Instances, and Single Instances.

**Evaluation Metric.** We evaluated the performance at both the image-level and the pixel-level, using the Area Under the Receiver Operating Characteristic curve (AUROC) as the primary metric for quantifying image-level (I-AUC) and pixel-level (P-AUC) performance. To ensure a more equitable treatment of anomaly regions of varying sizes, we employed the Per-Region-Overlap (PRO) metric for anomaly segmentation.

### 4.2 IMPLEMENTATION DETAILS

We pre-train the teacher model using the pre-trained WideResnet101 Zagoruyko & Komodakis (2016) on the ImageNet dataset. Both the teacher and student models use the small version of the Patch Description Network (PDN), with the student model's output feature dimension twice that of the teacher model, and the autoencoder encodes and decodes the complete image through a bottleneck of 64 latent dimensions. Our hyperparameter settings are as follows. $\lambda_{t-s}$, $\lambda_{t-ae}$, $\lambda_{ae-s}$ are all set to 1. During both training and fine-tuning, the teacher model is frozen. The Adam optimizer is used with a learning rate of 0.00001 for the student model and the autoencoder. Experiments are conducted by default on an NVIDIA Geforce GTX 3090Ti with 24 GB of RAM. We train our model for 70k iterations, with a maximum of 60k iterations for fine-tuning training. However, our experiments show that the model often achieves the best performance before reaching the full 60,000 iterations. We compared our quantization method with HQS Esser et al. (2019) and OMPQ Ma et al. (2023). We fix the weight and activation of the first and last layer at 8 bits, following previous works, where the search space is 2/3/4/8 bits. Note that during comparative experiments, we disabled OMPQ's mixed precision method.

### 4.3 MAIN RESULTS

We choose the architecture of EfficientAD Batzner et al. (2024) as our benchmark for evaluating our method. EfficientAD is an unsupervised IAD method utilizing a lightweight feature extractor, achieving both low error rates and high computational efficiency. We believe that the trade-off between accuracy and speed is the direction for future IAD development. It is worth noting that EfficientAD has not publicly released its code, so we reproduced their method and refer to it as baseline[†] in Table 1.

Figure 6: We conducted evaluations across three datasets, and use "TRUE"/"FALSE" evaluation metrics. "TRUE": Proportion of correct predictions in the predicted mask $M_{pred}$ relative to the ground truth (GT). "FALSE": Proportion of erroneous predictions in $M_{pred}$ relative to itself.

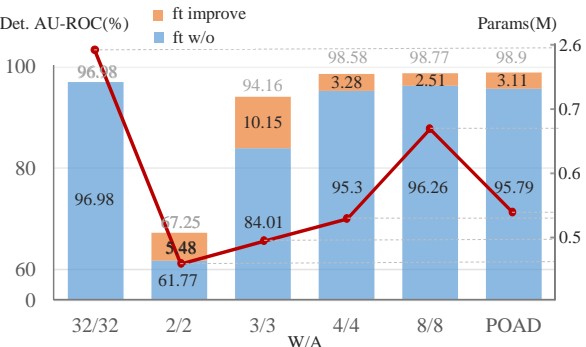

Figure 7: Quantizing and fine-tuning models using varying bit-width. Constructing stacked bar charts with I-AUC corresponds to the left. Plotting line graphs with model parameters corresponding to the right.

| Dataset | Baseline | RAAD |
|---------|----------|------|
| MVTec | 87.8 / 35.1 | 89.2 / 33.4 |
| LOCO | 63.1 / 55.8 | 66.0 / 52.9 |
| VisA | 93.3 / 75.3 | 93.6 / 75.2 |

To demonstrate the improvement of RAAD on the model's anomaly detection performance, we conducted extensive experiments on 32 datasets across three IAD datasets. We applied various post-training quantization methods to the original framework for comparison, with all quantization methods performing maximum quantization, i.e., both weights and activations are quantized to 8 bits, ensuring the bit-width is greater than or equal to that used by RAAD quantization. As shown in Table 1, RAAD achieved average I-AUC scores of 98.8, 89.75, and 96.13 on MVTec-AD, MVTec-LOCO, and VisA, respectively. "LSQ" and "OMPQ" denote the results when our method's quantization is replaced with the quantization methods from Esser et al. (2019) and Ma et al. (2023), respectively, both of which are representative post-training quantization methods. RAAD consistently outperforms baselines, emphasizing its effectiveness in both image-level anomaly detection and pixel-level anomaly localization. This demonstrates the framework's adaptability to diverse anomaly characteristics.

In addition, we also compared RAAD with several competitive methods across multiple datasets using various evaluation metrics, as shown in Table 2. We compare RAAD with PatchCore Roth et al. (2022), GCAD Bergmann et al. (2022), and SimpleNet Liu et al. (2023). Besides EfficientAD, the results of the other methods are from "paper with code". RAAD's average I-AUC score across the three datasets is 95.12, which is 5.37, 7.95, 6.75, and 2.97 higher than the other methods, demonstrating the best overall anomaly detection performance, proving RAAD achieves powerful image-level detection and pixel-level anomaly localization.

## 4.4 EMPIRICAL STUDIES

**Effectiveness of different components.** We investigate the effectiveness of each component of RAAD in Table 3. We set the baseline as the performance of EfficientAD-S on MVTec AD, comparing the Mean I-AUC and Mean PRO. Utilizing post-training quantization leads to improvements of 1.79% and 0.79%, respectively. Incorporating Hierarchical Quantization Scoring (HQS) results in improvements of 0.13% and 0.75% respectively, indicating that the introduction of HQS can further enhance the model performance. Quantization alone improves performance by mitigating attention bias, but HQS significantly optimizes attention distribution, emphasizing its critical role in refining anomaly sensitivity.

**Prove the quantization effect.** In order to better prove the ability of quantification to solve problems, we added two metrics. In Table 6, we evaluate three benchmark datasets (MVTec, VisA, and LOCO). The results of "Vanilla" (original EfficientAD) and "RAAD" (our method) are reported as all class averaged values. "TRUE": Proportion of correct predictions in the predicted mask $M_{pred}$ relative to the ground truth (GT), formulated as: $TRUE = \frac{M_{pred} \bigcap GT}{GT}$. A higher TRUE indicates better alignment between predictions and GT. "FALSE": Proportion of erroneous predictions in $M_{pred}$ relative to itself, formulated as: $FALSE = \frac{M_{pred} - M_{pred} \bigcap GT}{M_{pred}}$. A lower FALSE signifies higher precision in predictions. Analysis table, TRUE: In MVTec/VisA, Minimal differences before/after

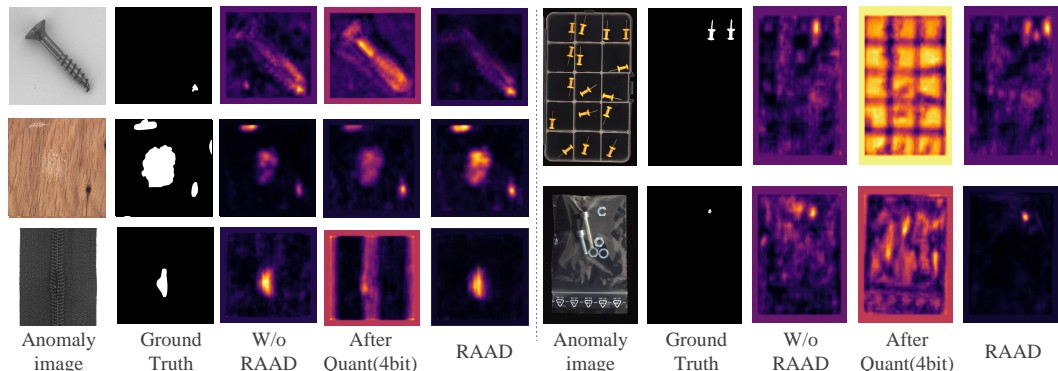

Figure 8: Qualitative results of RAAD on the MVTec-AD dataset Bergmann et al. (2019a) and MVTec-AD LOCO Bergmann et al. (2022). Within each group, from left to right, are the abnormal image, true value, baseline method prediction, quantized model prediction, RAAD prediction.

quantization. In LOCO, Significant TRUE improvement after quantization, attributed to weaker baseline performance allowing larger GT coverage. Similar trends were observed in underperforming MVTec/VisA classes. FALSE: In MVTec, increased FALSE post-quantization, indicating broader attention coverage. In VisA, Slight FALSE decrease due to poor baseline performance.

**Quantization and Hierarchical Quantization Scoring (HQS).** Figure 7 illustrates the effect of varying quantization bit-widths on model performance, including fine-tuning results for different configurations. Constructing stacked bar charts with I-AUC corresponds to the left vertical axis. Plotting line graphs with model parameters corresponding to the right vertical axis. It can be seen that wider bit-widths better preserve the original model performance, while lower bit-widths increase precision loss, posing challenges for fine-tuning to recover dimensionality. However, it is also observed that fine-tuning is crucial for improving the performance of the quantized model, especially when using lower bit-width quantization, which can lead to more significant improvement. Notably, after applying the HQS method, where the quantization width does not exceed 8 bits, the performance before fine-tuning is slightly lower than that of 8-bit quantization. However, after fine-tuning, it surpasses 8-bit quantization, demonstrating that a more suitable bit-width can better preserve key correspondences. At the same time, the line chart in Figure 7 visualizes the model parameters. Compared to other results with fixed bit widths, RAAD achieves higher accuracy with fewer parameters.

**Qualitative Results** Figure 8 presents the qualitative results of RAAD on the MVTec-AD and MVTec LOCO datasets. We have visualized the anomaly maps at different stages. Within each group, from left to right, are the anomaly image, ground-truth, predicted anomaly score from EfficientAD-S Batzner et al. (2024), predicted anomaly score from after model quantization (with the quantization bit set to 4-bit to highlight the differences), and the anomaly maps generated by RAAD. It is evident that the anomaly maps after quantization exhibit significant diffusion. The anomaly maps produced by RAAD are more accurate than the baseline, with lower anomaly probabilities in the normal regions.

## 5 CONCLUSION

Unsupervised IAD methods generally suffer from intrinsic bias in normal samples, which results in misaligned attention. This bias causes models to focus on variable regions while overlooking potential defects in invariant areas. In response, to this work, we propose RAAD (Recalibrating Attention of Industrial Anomaly Detection), a comprehensive framework that decomposes and recalibrates attention maps through a two-stage quantization process. By employing the Hierarchical Quantization Scoring (HQS) mechanism, RAAD optimally redistributes computational resources to enhance defect sensitivity. Qualitative and quantitative experiments show that our method can allocate the model attention properly, breaking the bias of unsupervised IAD, and achieving effective attention redistribution.

## 6 ETHICS STATEMENT

This work adheres to the ICLR Code of Ethics. In this study, no human subjects or animal experimentation was involved. All datasets used, including MVTec-ADBergmann et al. (2019a),MVTec-LOCO Bergmann et al. (2022) and VisA dataset Zou et al. (2022), were sourced in compliance with relevant usage guidelines, ensuring no violation of privacy. We have taken care to avoid any biases or discriminatory outcomes in our research process. No personally identifiable information was used, and no experiments were conducted that could raise privacy or security concerns. We are committed to maintaining transparency and integrity throughout the research process.

## 7 REPRODUCIBILITY STATEMENT

We have made every effort to ensure that the results presented in this paper are reproducible. All code and datasets have been made publicly available in an anonymous repository to facilitate replication and verification. The experimental setup, including training steps, model configurations, and hardware details, is described in detail in the paper. We have also provided a full description of HQS, to assist others in reproducing our experiments.

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

## A  INFERENCE SPEED, GPU MEMORY, AND MODEL PARAMETERS

Table 4: Comparing different PTQ methods and bit widths on the inference speed, GPU memory, and model parameters.

| Method | W/A | FPS | MB | Params |
|---|---|---|---|---|
| baseline | 32/32 | 167 | 30.70 | 8.05 |
| LSQ | 2/2 | 147 | 30 | 0.50 |
| | 3/3 | 147 | 30 | 0.75 |
| | 4/4 | 147 | 30 | 1.00 |
| | 8/8 | 147 | 30 | 2.01 |
| OMPQ | 2/2 | 112 | 61 | 1.32 |
| | 3/3 | 112 | 61 | 1.20 |
| | 4/4 | 112 | 61 | 1.46 |
| | 8/8 | 112 | 61 | 1.74 |
| RAAD | 2/2 | 113 | 61 | 1.32 |
| | 3/3 | 113 | 61 | 1.20 |
| | 4/4 | 113 | 61 | 1.46 |
| | 8/8 | 113 | 61 | 1.74 |

Table 5: Comparison of existing advanced IAD methods and RAAD in inference time, memory usage, and model parameters.

| Method | FPS | MEM(MB) | Params(M) |
|---|---|---|---|
| EfficientAD-S | 167 | 30.7 | 8.05 |
| PatchCore | 25 | 267 | 68 |
| RD4D | 6 | 18 | 150 |
| SimpleNet | 4 | 8 | 72 |
| CPR | 125 | 3226 | 2.88 |
| RAAD | 113 | 61 | 1.46 |

Table 6: The advanced IAD method uses detection strategies and image-level detection accuracy I-AUC and pixel-level accuracy PRO.

| Method | strategy | AU-ROC | AU-PRO |
|---|---|---|---|
| EfficientAD-S | T-S+AutoE | 96.98 | 91.38 |
| PatchCore | Memory Bank | 99 | 93.5 |
| RD4D | T-S | 98.5 | 93.9 |
| SimpleNet | AD Synthesis | 98.1 | 92.9 |
| CPR | Memory Bank | 99.7 | 97.8 |
| RAAD | T-S+AutoE | 98.9 | 92.92 |

In Table 4, the impact of different post-training quantization methods and bit widths on the Inference Speed, GPU Memory, and Model Parameters is demonstrated.

In Table 5 and Table 6, a comparison of the existing state-of-the-art IAD methods with RAAD in terms of model size and inference time is presented, along with the detection strategies used by these methods. We believe that industrial defect methods suitable for edge devices should not only

have a low false negatives rate and false positives rate but also possess three key attributes: 1. Fast inference speed (high FPS), 2. Low GPU memory, and 3. Small model parameters. In Table 5, it can be observed that the memory bank strategy using feature embeddings achieves the best results in image-level detection but also incurs a certain memory overhead. Reconstruction-based strategies perform better at the pixel level.

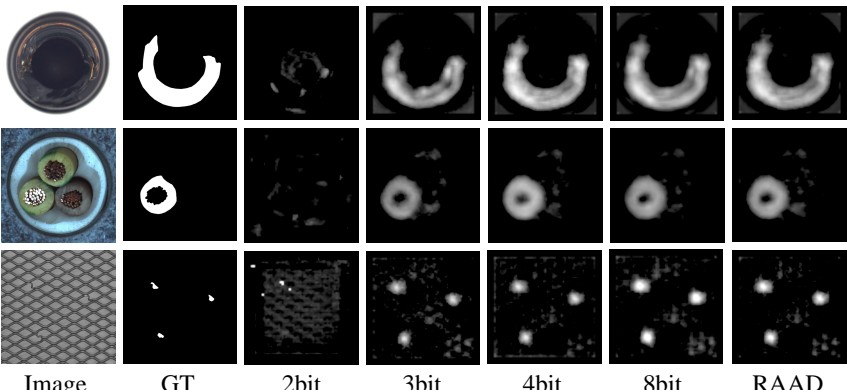

| Image | GT | 2bit | 3bit | 4bit | 8bit | RAAD |

Figure 9: Visualization of anomaly maps on MVTec AD with baseline and different PTQ methods.

## B  ADDITIONAL EXPERIMENTS

### B.1  COMPARISON WITH MODELS USING TRANSFORMER ARCHITECTURE

We compared the image-level AU-ROC metrics on the MVTec-AD dataset with methods using Transformer architectures. It can be observed that Transformer-based methods show significant performance variation across different categories, whereas RAAD is more balanced. This demonstrates that RAAD has stronger adaptability and generalization across different categories, and its overall score is superior to other methods.

We also tested our method based on the FOD approach. As shown in Table 7, FOD-Quant represents the quantized FOD model. It is important to clarify that our method is based on a CNN model, and the attention mentioned in the text is an analogy, not equivalent to the attention mechanism in Transformers.

### B.2  USING DIFFERENT NETWORKS

In Table 8, the impact of replacing the RAAD feature extraction network PDN with Wide-ResNet of different depths on model accuracy, model size, and inference time is demonstrated. Since the depth of Wide-ResNet needs to satisfy $(depth - 4)\%6 = 0$, we choose depths of 28, 40, 64, and 100.

### B.3  FULL RESULT OF THE VISA.

We provide the complete results on the VisA datasets in Table 9. As shown in the Table, we evaluated the anomaly detection presented in the image-level AU-ROC, and pixel-level AP.

## C  QUALITATIVE RESULTS

We visualized the results on the MVTec-AD dataset, as shown in Figure 9, demonstrating the impact of our method using different quantization bit widths on anomaly maps. Moreover, in Figure 10, we compare the different PTQ methods on MVTec AD and MVTec LOCO-AD. The baseline results are obtained using EfficientAD-S Batzner et al. (2024).

Table 7: Compared to the Image-level AU-ROC metric on the MVTec-AD dataset using Transformer architecture methods, "FOD-Quant": Quantitative evaluation results of the FOD.

| Category | | InTra (CVPR 2021) | UniAD (NeurIPS 2022) | FOD (ICCV 2023) | FOD (Quant) | RAAD |
|---|---|---|---|---|---|---|
| textures | Carpet | 98.8 | 98.5 | 99.6 | 97.9 | 98.7 |
| | Grid | 100 | 96.5 | 99.6 | 99.4 | 99.8 |
| | Leather | 100 | 98.8 | 100 | 99.8 | 98.3 |
| | Tile | 98.2 | 91.8 | 100 | 99.9 | 100 |
| | Wood | 97.5 | 93.2 | 98.8 | 98.2 | 98.5 |
| | average | 98.9 | 95.7 | 99.6 | 99.1 | 99.0 |
| object | Bottle | 100 | 98.1 | 100 | 99.8 | 100 |
| | Cable | 70.3 | 97.3 | 98.4 | 98.5 | 97.7 |
| | Capsule | 86.5 | 98.5 | 95.45 | 95.4 | 97.4 |
| | Hazelnut | 95.7 | 98.1 | 100 | 100 | 99.7 |
| | Metal_nut | 96.9 | 94.8 | 99.9 | 98.1 | 98.8 |
| | Pill | 90.2 | 95 | 94.2 | 95.0 | 98.0 |
| | Screw | 95.7 | 98.3 | 94.7 | 90.7 | 98.5 |
| | Toothbrush | 100 | 98.4 | 95.0 | 93.3 | 100 |
| | Transistor | 95.7 | 97.9 | 99.75 | 99.5 | 100 |
| | Zipper | 99.4 | 96.8 | 94.0 | 98.7 | 97.8 |
| | average | 93.0 | 97.3 | 97.1 | 96.9 | 98.8 |
| average all category | | 95.0 | 96.8 | 97.9 | 92.9 | 98.9 |

Table 8: Using Wide-ResNet of different depths on Inference FPS, GPU Memory, model parameters. And evaluate Detection AU-ROC and Segmentation AU-PRO in the capsule category. The units of Mem and params are MB and M, respectively.

| Depth | 28 | 40 | 64 | 100 |
|---|---|---|---|---|
| FPS | 178 | 125 | 91 | 65 |
| Mem | 9.6 | 11 | 14 | 18 |
| Params | 2.5 | 2.8 | 3.6 | 4.8 |
| I-AUC | 89.1 | 98 | 92.3 | 80.9 |
| PRO | 85.1 | 91.3 | 87.8 | 83.9 |

Table 9: The full result in the VisA dataset. We report the I-AUC and I-AP.

| VisA | Baseline | RAAD |
|---|---|---|
| candle | 91.72 / 81.90 | 92.92 / 87.50 |
| capsules | 85.15 / 76.03 | 85.13 / 79.46 |
| cashew | 86.15 / 76.03 | 98.14 / 95.05 |
| chewinggum | 98.68 / 96.08 | 99.64 / 98.99 |
| fryum | 97.72 / 95.88 | 98.61 / 96.91 |
| macaroni1 | 96.54 / 87.85 | 98.48 / 86.84 |
| macaroni2 | 89.26 / 78.26 | 92.05 / 82.57 |
| pcb1 | 99.17 / 96.08 | 99.53 / 97.98 |
| pcb2 | 98.94 / 95.92 | 99.28 / 96.08 |
| pcb3 | 95.71 / 89.11 | 97.56 / 92.93 |
| pcb4 | 99.35 / 94.34 | 99.60 / 96.15 |
| pipe_fryum | 99.54 / 98.00 | 99.72 / 99.00 |
| average | 94.73 / 88.79 | 96.72 / 92.05 |

## D    ALGORITHM

The following algorithm explains and proves the effectiveness of quantization and fine-tuning.

For example, once an iteration of fine-tuning:

$$\hat{W}_1 = W^+ - \eta \nabla L(W^+) = W^* + \Delta W - \eta H(W^*)\Delta W \Leftrightarrow \hat{W} - W^* = (I - \eta H(W^*))\Delta W \quad (7)$$

Next, we explain why $(I - \eta H(W^*))\Delta W$ is better than $H(W^*)\Delta W$. We decompose $H(W^*)$ into its eigenvalues as: $H(W^*)\Delta W = U\Lambda U^T$. 1. When $U$ is large, the model is currently at a sharp

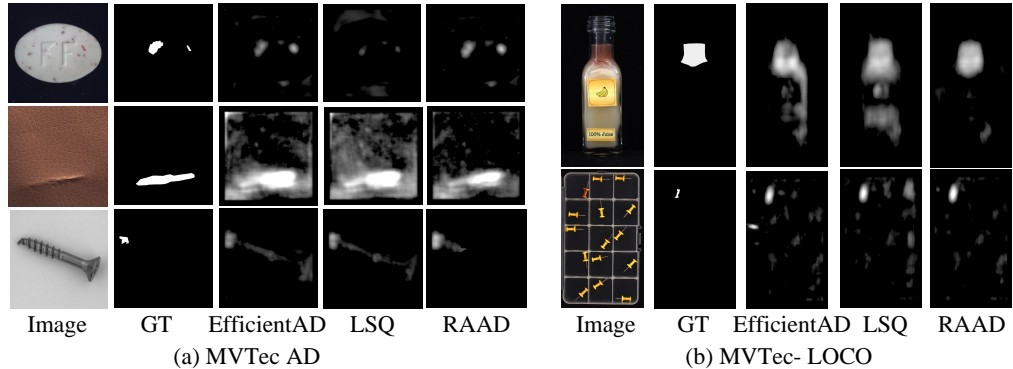

| Image | GT | EfficientAD | LSQ | RAAD | Image | GT | EfficientAD | LSQ | RAAD |

(a) MVTec AD                (b) MVTec- LOCO

Figure 10: Visualization of anomaly maps on MVTec AD and MVTec-LOCO.

---

**Algorithm 1** Proof of Regularization through Quantification

---

**Input:** the initial model parameters $W$ ; after training $W^*$; quantization model parameters $W^+$;
     learning step $\eta$; training loss $L(\cdot)$; quantization $Quant(\cdot)$;
1: $W^* = arg \min_W L(W)$, training convergence $\bigtriangledown L(W^*) = 0$;
2: $W^+ = Quant(W^*) = W^* + \Delta W$;
3: $\hat{W} = W^+ - \eta \bigtriangledown L(W^+) \Leftrightarrow \hat{W} - W^+ = -\eta \bigtriangledown L(W^+)$;
4: $\bigtriangledown L(W^+) \xrightarrow{2} \bigtriangledown L(W^* + \triangle W)$;
5: using Taylor expansion, $\bigtriangledown L(W^*) + H(W^*) \triangle W + R(\triangle W) \xrightarrow{1} H(W^*) \triangle W + R(\triangle W) \approx$
     $H(W^*) \triangle W$;
6: $\because H(W^*)\Delta W \neq 0 \therefore$ After quantization-fine tuning, the gradient is no longer 0.

---

minimum. $I - \eta H(W^*)$ facilitates escaping from this local minimum. After multiple iterations, the model will tend to approach regions with predominantly smaller Hessian eigenvalues, known as flat minima. 2. When $U$ is small, indicating that the model is at a flat minimum, $I - \eta H(W^*) \approx I$, and the model performance remains stable.

It is well known that the test loss $L_{test}$ is equal to the training loss $L_{train}$ plus the generalization ability $L_{gene}$. When a model overfits the training data by learning specific details and noise, it loses generalization ability, resulting in a high $L_{test}$ despite a low $L_{train}$. This occurs because the model has learned features specific to the training data that do not apply to new, unseen data. Quantization, as a regularization technique, enhances model generalization and achieves optimization of model performance during fine-tuning.

## E    FUTURE WORKS

Unlike model compression, our approach employs model quantization not with the aim of reducing model size, but rather to achieve dimensionality reduction at the precision level. Some industrial anomaly detection efforts currently utilize model quantization during deployment to reduce model size while maintaining certain accuracy levels. For instance, CPR Li et al. (2023) has achieved 1016 FPS using TensorRT on NVIDIA RTX4090 GPU.

In future work, we intend to propose a post-training quantization method specifically designed for the industrial anomaly detection domain, utilizing PyTorch. This method would be applied to post-training models to reduce model size while maintaining or even enhancing model accuracy.

Additionally, we aim to conduct experiments on the more challenging IAD dataset. Currently, most state-of-the-art methods have reached saturation (AUROC exceeding 99%) on mainstream datasets like MVTec, making it difficult to distinguish between methods, which leads to unsatisfactory performance in practical applications. A recently introduced dataset, Real-IAD Wang et al. (2024), has garnered attention due to its large scale, real-world context, and multi-view nature. In future work, we plan to experiment with this dataset.

## F  LLM USAGE

Large Language Models (LLMs) were used to aid in the writing and polishing of the manuscript. Specifically, we used an LLM to assist in refining the language, improving readability, and ensuring clarity in various sections of the paper. The model helped with tasks such as sentence rephrasing, grammar checking, and enhancing the overall flow of the text.

It is important to note that the LLM was not involved in the ideation, research methodology, or experimental design. All research concepts, ideas, and analyses were developed and conducted by the authors. The contributions of the LLM were solely focused on improving the linguistic quality of the paper, with no involvement in the scientific content or data analysis.

The authors take full responsibility for the content of the manuscript, including any text generated or polished by the LLM. We have ensured that the LLM-generated text adheres to ethical guidelines and does not contribute to plagiarism or scientific misconduct.

