# OpenReview forum: "Breaking the Bias: Quantifying the Attention of Industrial Anomaly Detection"
_ICLR.cc/2026/Conference — ICLR 2026 Conference Withdrawn Submission_

### Official Review · Reviewer_NiTp · 2025-10-18

**Soundness:** 3
**Presentation:** 2
**Contribution:** 2
**Rating:** 4
**Confidence:** 4

**Summary:**

This paper addresses the task of image anomaly detection. Motivated by the observation that the feature map attention of existing models often focuses on semantically variable regions, this work proposes two modules to recalibrate this attention. Specifically, our method modifies attention maps through quantization and subsequent fine-tuning. This approach readjusts the model's attention, enabling it to achieve competitive performance on standard benchmarks while maintaining high efficiency.

**Strengths:**

The authors introduce a novel quantization-aware training framework designed to recalibrate the attention maps of deep learning models, thereby enhancing their performance on anomaly detection tasks.

The approach is both effective and efficient, demonstrating competitive results across multiple benchmarks.

The core ideas are presented clearly, though the paper would be strengthened by polishing the language to fix occasional obvious errors and improve the clarity of certain claims.

**Weaknesses:**

1.  Regarding the motivation for the model's bias towards normal samples: While Figure 1 provides some examples, the concept remains unclear.
    * How is this bias quantitatively measured? Is it the average magnitude of the difference between the student and teacher networks?
    * Regarding Figure 1, it is not specified how the feature map was generated or from which baseline model it originates.
    * Is this bias a general phenomenon across different anomaly detection models, or is it unique to the baseline model studied in this work? Clarifying these points would strengthen the paper's motivation.

2.  Regarding the proposed quantization-aware training:
    * It is unclear why this method leads to better performance compared to a network with higher-precision parameters. Please clarify the underlying mechanism.
    * The comparison between the proposed method and the baseline may be unfair. Since quantization-aware training is used as a fine-tuning stage, a fairer comparison would involve training the baseline for a comparable number of total steps.
    * While Figure 7 presents a comparison across different quantization settings, it is missing the result for the unquantized (32/32) setting after fine-tuning, which would serve as a crucial control.

3.  Regarding the baseline in Table 3: Lines 373-374 state that the proposed method leverages EfficientAD as the baseline. Does the "vanilla" setting in Table 3 refer to the reported results from the original EfficientAD model?

4.  Regarding Section 3.4 (Attention Map Recalibration): The title of this section is potentially misleading. The content and equations seem to describe how the model is supervised rather than a method for calibrating the attention map itself. Please clarify this discrepancy.

5.  The manuscript requires several corrections to the text and formatting:
    * Lines 189-190: The notation '$E$' is used without being previously defined.
    * Lines 229-230: A key equation for the weighted objective appears to be missing, which makes the method difficult to fully understand.
    * Line 303: The subscript '$h$' is missed in '${D}$'.
    * References: The citation format appears incorrect. Please ensure all citations are enclosed in parentheses, e.g., (Author, Year), as is standard.

**Questions:**

Please see the weakness section for my main concerns.

---

### Official Review · Reviewer_HvPx · 2025-10-26

**Soundness:** 3
**Presentation:** 1
**Contribution:** 1
**Rating:** 2
**Confidence:** 4

**Summary:**

The paper builds upon the line of knowledge distillation methods for anomaly detection, where a student network learns to distill the knowledge of a teacher network using only normal samples during training. At inference time, the discrepancy between the teacher and student outputs is used to identify abnormal inputs. The main novelty of this work lies in quantizing the anomaly maps derived from the teacher–student differences to suppress irrelevant high-value regions while preserving the main areas of true discrepancy.

**Strengths:**

1 - The idea of using quantization to filter out the noise on discrepancy maps is sound and interesting.

2 - There are enough visualizations in the paper showing the effectiveness of applying quantization.

**Weaknesses:**

1. The method adopts EfficientAD as the baseline and iteratively introduces modifications; however, the reported performance is consistently below the original paper’s results (MVTecAD: 98.8 → 96.9, LOCO: 90.0 → 84.0, VISA: 97.5 → 95.3). Even the final variant (RAAD) underperforms the baseline as reported in the literature, making it difficult to assess the effectiveness of the proposed changes.

2. The method description feels disjointed. For example, the paper presents Stage 1 and then introduces the quantization formula without clearly specifying how Stage-1 outputs feed into Stage-2. Similarly, the autoencoder component appears abruptly, with limited justification or background on its role and integration in the pipeline.

3. Prior work uses various mechanisms to suppress irrelevant regions (e.g., bottleneck compression); in some cases, a simple Gaussian filter effectively reduces spurious high responses. Including such straightforward baselines in the comparison would clarify the incremental value of the proposed quantization.

4. The set of SOTA baselines is outdated. More recent methods should be added to the comparison tables to reflect the current state of the field[1,2,3].

5. Table 2 is inconsistent: Pixel-AUROC is missing for two datasets despite being mentioned in the caption. Please complete or correct the table for consistency.

[1] - GeneralAD: Anomaly Detection Across Domains by Attending to Distorted Features, ECCV24

[2] - Dinomaly: The Less is More Philosophy in Multi-Class Unsupervised Anomaly Detection, CVPR2025

[3] - AnomalyDINO: Boosting Patch-based Few-shot Anomaly Detection with DINOv2, WACV25

**Questions:**

Please refer to the weaknesses.

---

### Official Review · Reviewer_3NGY · 2025-10-28

**Soundness:** 1
**Presentation:** 1
**Contribution:** 2
**Rating:** 2
**Confidence:** 4

**Summary:**

The article is about industrial visual quality control by unsupervised anomaly detection. Many recent articles have been reaching high anomaly detection performances by using deep learning. The utilization of neural networks for this kind of problems is challenging, as neural networks require supervision, in the form of a labelized dataset, and anomalies are generally considered to be unknown and unavailable for training.

This article proposes an improvement of the Student-teacher (ST) framework, specifically of the EfficientAD method. The ST methods use asymmetric knowledge and overfitting as anomaly detection mechanisms. More specifically, a teacher network is pre-trained to have a global knowledge/representation of natural images, and a student network is trained to imitate and overfit the teacher, but only on the normal images (inliers). It is supposed that the student’s representation will diverge from the teacher's representation on anomalies (outliers).

The proposed model first trains three networks, like EfficientAD, namely a Teacher, a student and an Auto-Encoder. The Auto-Encoder is an additional student with a different architecture, compressing and decompressing the information, and thus informing on global or logical anomalies. Then the models are hierarchically/adaptively quantized to gain efficiency and reduce the model bias. Finally the quantized models are fine-tuned again with the original student teacher loss, to correct their alignment.

The experiment uses three well known Industrial anomaly detection benchmark datasets (MvtecAD, MvtecLoco and Visa). The authors compare the method detection performances at image and pixel level with the state of the art methods. They also conduct a detailed ablation study.

**Strengths:**

Most unsupervised normality models used a pre-trained network for image embedding. However, few study the impact of the network choice, or of the layer choice. In a study[1] from the authors of the MvTec datasets, it is shown that indeed the choice of the network and the layer has a big impact on the performances, but that there is no clear way to choose the best design without brute force testing.

So I think that studying this subject and designing an automatic solution for weighting the importance of each layers in the image embedding is important and has a potential impact for all the field.

[1] Lars Heckler, Rebecca König, Paul Bergmann; Proceedings of the IEEE/CVF Conference on Computer Vision and Pattern Recognition (CVPR) Workshops, 2023, pp. 2917-2926

**Weaknesses:**

**Problem in the “bias problem” denomination:**


I don’t think that the baseline ST model’s errors are coming from a bias. Indeed, ST methods use mean square error (MSE) between the Teacher and the Students as a scoring function. Using the MSE is like supposing that the Student prediction on normal objects is a Gaussian random variable, centered on the Teacher’s prediction, and that this Gaussian is isotrope, i.e. that there is no correlation and the variance of the error is the same in all the directions. Directions here being all the layers, positions and channels.

Overfitting, on an unsupervised normality model will generate many false detection. As the model fits tightly the training data, any slight difference will trigger a detection. Therefore, for me, the problem is not bias or overfitting, but an over simple normality model. This anisotropy, in fact, is shown in the second column of Figure 1, because the mean of the score (MSE) is basically the variance, which is very localization dependent. This is visible because in these dataset the objects are manually registered so every pixel approximately shows the same part of the object.

Conversely, if the score was normalized by the variance of the normal scores, overfitting may arise if the normalizing constants ( the variances) are not correctly regularized.

For me there, the quantization has an impact because it is adaptive, i.e. it projects all the different scales ranges r^l into a unique range q, which can be seen as a kind of normalization. For me this is what shows Table 3, the big improvement is with quantization and between HQS and quantization the difference is thin. Same in table 1, the difference between OMPQ and RAAD is small. However the only problem that is solved is the layer anisotropy, not the localisation or channel anisotropy.

Also, the authors regularly mention “attention bias”, but these methods don’t use any attention mechanism. They output heatmaps and either threshold them for localization or take the max for image level detection. So, for me, this “attention” term is misleading. It is even more the case nowadays because the attention is generally used to speak about transformers.

So I think that the article's analysis on ST baselines limitation is erroneous. The authors can defend their interpretation  and solution, or show me where I am mistaken.

 **Originality**

The article is using the base anomaly detector method EfficientAD[2], from which comes the Loss (1) , (5) and (6). So it’s basically two EfficientAD training, and a quantization which is in fact the quantization method BRECQ[3], in equation (2) and (3). So the original work is the automatic quantization factor selection, in equation (4), which, as mentioned previously, seems to have a limited impact. There is also the discussion about the bias, from which I the explanation needs to be strengthened. I would like for the authors to argue in what way they work is a sufficiently original and important work.

**Also :**

[line 190] a feature extraction operator $E$ is introduced, but is not used in equation (1) or others. I guess that it is used in $L_{pre}$ to distillate WRN-101 into PDN, like in [2]. This should be either specified, suppressed or replaced by a citation. Or is $E$ and $\Psi$ the same thing ?

[Figure 5] The heatmap’s colormap and the colorbar’s colormap don’t match.
[Line 256] It is mentioned that the detailed explanation of how the piecewise constant function psi converts ST error into bit-wise is in the appendix, but I can’t find it.

[Table 1] Some results that are not the best of their line are in bold. Like in MvTec-AD average line the Image-level AP% of OMPQ is higher but RAAD is in bold.

[Figure 6][Line 427] What is called TRUE is in fact what is normally called Recall in machine learning, and what is called FALSE is one minus the precision. I think the authors should use the standard metrics, which will help the reading.

[Line 806, equation 7] I think $\hat{W}_1$ should just be $\hat{W}$. Also the notation $W^+$ and $W^{\star}$ should be explained. I guess that $W^+$ is the current quantized weight and $W^\star$ was the optimal weight before quantization and fine-tuning and $\Delta W$ is the difference between optimal and current quantized weight. The explanation is not very clear, I don’t understand in which scenario I will not have the identity matrix. Why is a sharp minima necessarily worse than a flat one ?

[Line 809] I think in the eigen decomposition formula of the hessian, should not contain $\Delta W$.


[2] Batzner, K., Heckler, L., & König, R. (2024). Efficientad: Accurate visual anomaly detection at millisecond-level latencies. In Proceedings of the IEEE/CVF winter conference on applications of computer vision (pp. 128-138).
[3] Li, Y., Gong, R., Tan, X., Yang, Y., Hu, P., Zhang, Q., ... & Gu, S. (2021). Brecq: Pushing the limit of post-training quantization by block reconstruction. arXiv preprint arXiv:2102.05426.

**Questions:**

[Line 362] having a learning rate of 1e-5 is low compared to the more current 1e-3 or 1e-4. Is there any stability problem?

[Table 5 and 6] I think these two tables should be fuzed to make it easy to compare the trade off between efficiency and accuracy. Also in [Line 704] the detection results are coming from Table 6.

[Table 5] As I understand it RAAD is a quantization of EfficientAD ( the baseline). So I don’t understand how a quantized method can have lower FPS and higher memory consumption while having less parameters.

**Details Of Ethics Concerns:**

no ethical concerns

---

### Official Review · Reviewer_Zh4b · 2025-11-01

**Soundness:** 4
**Presentation:** 4
**Contribution:** 3
**Rating:** 8
**Confidence:** 3

**Summary:**

Paper proposes a novel method for the semi-supervised Industrial Anomaly Detection (IAD) problem. The proposed architecture consists of 3 components: teacher student model + autoencoder; During training and fine-tuning, only normal images are used. Initial
 training, followed by a novel hierarchical quantitative scoring (HSQ), and a final fine-tuning of attention. The HSQ aligns the teacher and student models and computes the anomaly scores are calculated using the outputs of the aligned convoluted layers. The anomaly scores guides the reallocation of attention to the defect-prone regions. Layers with higher scores are assigned with greater numerical precision in the quantization, and compress the layers which are "less informative" about the defects. A novel aggregated loss was also used to recalibrate the attention to minimise the overall differences of the respective recovered images.

**Strengths:**

Originality: Very high. The use of quantization for attention redistribution is novel for the IAD. Unlike prior works in quantization for anomaly detection which main objective is for model size and computation efficiency, the main focus of this work is to distribute the computation precision according to the "information gained" in the corresponding layers of the teacher-student pairs.

Quality: High. Hypothesis is well supported by the architectural and training design. The experimental setup and results are also good evidence of the central claim that the quantization method is suitable for the distribution of computing precision to "focus" on the layers with more "anomalous information".

Clarity: Excellent. Paper is well-written and method is clearly explained. Experimental results are well presented and supports the main claims of the paper.

Significance: Good. Paper borrows several new ideas from prior works and applies them well in the IAD. While there is no significant breakthrough in terms of architecture design, or other quantization method. The HSQ is a creative and novel application of quantization.

**Weaknesses:**

Experimental results are mixed to draw very strong conclusions about the validity of the claim that the distribution is significantly superior to prior SOTAs.

(minor) The proof-reading and formatting of the paper can be improved.
Figure 3. "Nominal images" should be "Normal images". Nominal's meaning is "(of a role or status) existing in name only", "(of a price or charge) very small; far below the real value or cost.".
Figure 3 and 4 should be combined.
Figure 5. The squeezing and position of the figure makes the reading of the text at line 216-230 difficult to read. It can be excluded, as Line 307 to 312 has explained the process clearly.
Table 1: Should be placed on Page 7 or after.

**Questions:**

Will the author please further elaborate and hypothesize on the "mixed results" of the proposed methods as compared to other SOTA (ref: Table 2), especially with regards to the MVTec dataset?

---

### Note · Authors · 2026-01-26

I have read and agree with the venue's withdrawal policy on behalf of myself and my co-authors.

---

### Meta-Review · Area_Chair_Qt3c · 2025-12-17

**Summary:**

**Reviewer Zh4b:**  1) The 'bias problem' considered in the paper hasn't been well motivated and analyzed. 2) The technical novelty is limited.

**Reviewer HvPx:** 1) The performance improvement of the proposed method over the baselines is not significant. 2) Some more relevant or recent baselines are not compared in the experiments.

**Reviewer NiTp:** 1) The 'bias problem' considered in the paper hasn't been clearly explained. 2) There is no concrete analysis of why the quantization is useful in improving the performance. 3) Many writing and formatting issues exist.

**Reviewer Concerns:**

Since the authors haven't provided any rebuttal, all of the aforementioned concerns should remain.

**Reviewer Scores:**

Reviewer Zh4b recommended "accept", while the other three reviewers opposed acceptance. The authors declined to respond to the comments of all reviewers.

I read the paper carefully and agree with the weaknesses pointed out by Reviewers Zh4b, Reviewer HvPx, and NiTp. The paper should be rejected because: 1) it failed to explain the "bias problem" theoretically and numerically; 2) the numerical improvement is tiny; 3) the usefulness of quantization in boosting anomaly detection hasn't been justified theoretically.

---

### Decision · Program_Chairs · 2026-01-26

Reject